# Type 2 diabetes remission and its predictors in an Indian cohort: A retrospective analysis of an intensive lifestyle intervention program

Pramod Tripathi[1,2], Nidhi Kadam[1]*, Thejas Kathrikolly[1], Diptika Tiwari[1], Anagha Vyawahare[1], Baby Sharma[1], Malhar Ganla[2], Banshi Saboo[3]

1 Department of Research, Freedom from Diabetes Research Foundation, Pune, Maharashtra, India, 2 Department of Management and Exercise Science, Freedom from Diabetes Clinic, Pune, Maharashtra, India, 3 Department of Medicine, Dia Care - Diabetes Care and Hormone Clinic, Ahmedabad, Gujarat, India

* research@freedomfromdiabetes.org

## Abstract

Predictors of type 2 diabetes (T2D) remission following intensive lifestyle intervention (ILI) are poorly characterized, especially in high-risk populations, such as India. This study aimed to identify the key predictors of T2D remission after an ILI in an Indian population. This retrospective analysis included 2384 patients with T2D (age 30–75 years; body mass index (BMI) ≥23 kg/m² enrolled in an online one-year ILI program at the Freedom from Diabetes Clinic, India, between May 2021 and August 2023. The intervention included personalized plant-based diet, physical activity, stress management, and medical support. Remission was defined as maintaining glycated hemoglobin (HbA1c) < 48 mmol/mol (6.5%) for ≥3 months without glucose-lowering medications. Anthropometric and biochemical data were extracted from clinical records. Predictors were assessed using logistic regression analysis. Post- intervention, 744 patients (31.2%) achieved remission The remission group showed significantly greater improvements in weight (−8.5% vs. −5.2%), BMI (−8.6% vs. −5.2%), HbA1c (−15.3% vs. −12.4%), fasting insulin (−26.6% vs. −11.4%), and homeostatic model assessment of insulin resistance (HOMA-IR) (−37.3% vs. −19.7%), than the non-remission group (p<0.05). The predictors of remission included age (≤50 years), higher BMI (≥25 kg/m²), drug-naïve status, shorter disease duration (≤6 years), juice fasting, baseline HbA1c<7%, weight loss >10%, and post-intervention HOMA-IR<2.5 (p <0.05). Our findings demonstrate that a significant proportion of individuals with T2D can achieve remission through a comprehensive culturally adapted lifestyle program. The identification of both baseline and post-intervention predictors underscores the importance of early, personalized, and holistic care in diabetes management.

**Data availability statement:** All relevant data are within the paper and its Supporting Information files.

**Funding:** The author(s) received no specific funding for this work.

**Competing interests:** The authors have declared that no competing interests exist.

## Introduction

Type 2 diabetes (T2D) is a global health concern, primarily driven by lifestyle factors and comorbidities. The International Diabetes Federation estimates that 537 million adults currently live with diabetes, which is projected to increase to 853 million by 2050 [1]. In India, cases have surged from 33 million in 2000 to 72 million in 2021, with projections of 125 million by 2045, posing a significant public health challenge [2]. Addressing this has become a critical priority for India's healthcare system. Given this alarming increase, T2D remission, defined as maintaining glycated hemoglobin (HbA1c) levels <48 mmol/mol (6.5%) for at least three months without glucose-lowering medications, is gaining importance [3].

There is growing interest in non-pharmacological interventions for achieving remission [4,5]. Non-pharmacological strategies, especially lifestyle interventions focusing on diet and weight management, are central to remission efforts [5,6]. A recent position statement by the American College of Lifestyle Medicine stated that remission rates of nearly 50% are achievable through lifestyle changes [7]. Trials have shown that Mediterranean, low-carbohydrate, ketogenic, and plant-based diets effectively lower HbA1c and improve remission rates [8]. Additionally, large-scale trials have emphasized the role of behavioral therapy in achieving T2D remission, supporting a multidisciplinary approach for diabetes management [9]. However, most studies have focused on single interventions, mostly low-calorie diets, with medication withdrawal [4,5]. Despite the success of these approaches in Western populations, there are limited data on their effectiveness in India, where genetic and lifestyle factors place the population at higher risk. Existing interventions may not be directly applicable to the Indian population because of differences in genetics, phenotypes, and cultures. There is a critical need for culturally sensitive and tailored interventions to improve remission rates and health outcomes in India. Furthermore, research identifying predictors of T2D remission in this population remains limited [10].

This study addressed this gap through a retrospective analysis of patients enrolled in an intensive lifestyle intervention program, providing valuable real-world insights into its effectiveness in a large and diverse Indian cohort. The novelty of the intervention lies in its multidisciplinary approach, integrating diet, exercise, psychological support, and medical management, all of which are adapted to the Indian context. This study also aimed to identify predictors of T2D remission to guide culturally appropriate diabetes management strategies.

## Materials and methods

### Study design and setting

This study was conducted at the Freedom from Diabetes clinic, which operates on a one-year online subscription-based diabetes management model. The complete records of 2384 individuals with T2D enrolled between May 2021 and August 2023 across 263 Indian cities who met the eligibility criteria, were extracted.

## Eligibility criteria

Participants aged 30–75 years with a confirmed diagnosis of T2D and on oral hypoglycemic agents (OHAs) were included in the study; those on external insulin therapy were excluded. Drug-naïve patients with HbA1c levels of ≥6.5% were also included. Additional inclusion criteria included a body mass index (BMI) of ≥23 kg/m² and the availability of data from at least three consultations over a one-year program duration. Exclusion criteria comprised individuals with pancreatitis or drug-induced diabetes (such as steroid-induced diabetes), and other types of diabetes (including type 1 diabetes mellitus, diabetes insipidus, maturity-onset diabetes of the young, latent autoimmune diabetes in adults, or gestational diabetes). Patients with advanced complications, such as nephropathy (eGFR<30 mL/min/1.73 m² or urine microalbumin >1000 mg), severe retinopathy (severe non-proliferative or proliferative diabetic retinopathy), significant neuropathy (including complete numbness or sensory loss, amputations, ulcers, foot deformities, or Charcot foot), or peripheral arterial disease, were also excluded. Individuals with a known history of cancer, pregnant or lactating women, those who had been hospitalized for diabetes-related complications in the past six months, and those with incomplete or missing data were also excluded.

The patient selection process is illustrated in Fig 1. Of the 7,458 participants initially enrolled, 3,997 were excluded due to fewer than three follow-up visits, indicating dropout or minimal engagement. An additional 1,077 participants were excluded owing to incomplete or missing data, resulting in a final sample of 2,384 participants.

## Measurements

Baseline and one-year follow-up data were obtained from the electronic medical records of the clinic. The data for the present analysis were accessed on September 14, 2024.

**Anthropometric and sociodemographic.** Data pertaining to sociodemographic variables (including age, sex, marital status, education, and occupation) and anthropometric measurements (specifically weight and height) were obtained

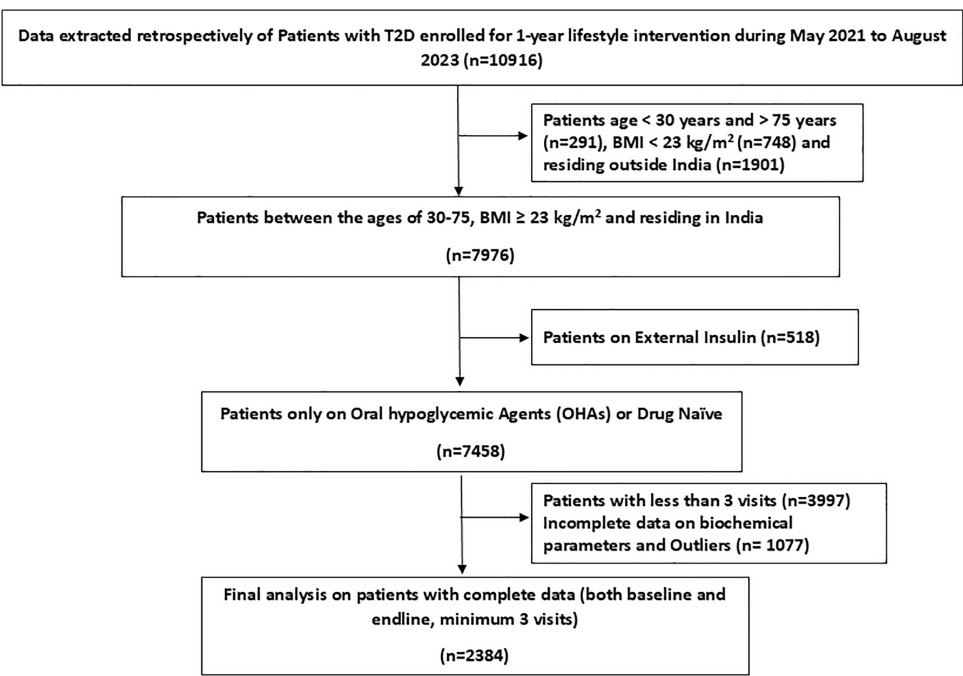

**Fig 1. Flowchart depicting the selection procedure of patients for the study.**

through self-reporting. To ensure accuracy, the participants received detailed instructions on the weight and height measurements. Weight was recorded as the average of three measurements, and BMI was calculated.

**Biochemical.** Data on HbA1c, fasting blood glucose (FBG), and fasting insulin were extracted from biochemical reports uploaded by participants and stored in a centralized data management software. Patients were asked to complete the tests at laboratories accredited by the National Accreditation Board for Testing and Calibration Laboratories (NABL), ensuring alignment with international HbA1c assay standards (e.g., National Glycohemoglobin Standardization Program (NGSP) and International Federation of Clinical Chemistry and Laboratory Medicine (IFCC)). The mobile application used for participant monitoring was integrated with the data management system, enabling real-time tracking and quality control of the submitted reports. Homeostatic Model Assessment of Insulin Resistance (HOMA-IR) and Homeostatic Model Assessment of β-cell function (HOMA-β) were calculated using standard formulae and analyzed in a subset of patients with available fasting insulin data (n = 1897) [11]. Remission was defined as maintaining HbA1c < 6.5% for at least three months without the use of glucose lowering medications in patients on OHAs and those who were drug-naïve [3]. As a program prerequisite, all participants were required to own a glucometer, and digital weighing scale for daily self-reporting of blood glucose level (BGL) and weight.

### Intervention

The one-year ILI was structured into four phases, each focusing on four core components: dietary modification, physical activity, psychological support, and medical management. The detailed protocol has been previously described [12].

Each participant was assigned to a six-member care team comprising a physician, dietitian, physical therapist, psychologist, mentor (a past program participant), and monitor (for follow-up and scheduling). At baseline, the core team (physician, dietitian, and physical therapist) set individualized health goals: the physician set HbA1c targets, the dietitian determined weight loss and BMI goals, and the physical therapist prescribed exercise targets considering any pre-existing conditions. These initial goals covered the first six months and were revised midway to establish advanced goals for the remaining six months.

Dietary intervention emphasized a plant-based diet (lentil-based recipes, sprouts, and vegetable salads) tailored to individual BMI and health conditions. Phase 1 emphasized a balanced alkaline diet rich in vegetables, antioxidants, and phytonutrients, with a caloric intake of 1200–1400 kcal/day. In Phase 2, intermittent fasting and juice fasting were introduced for all participants. Juice fasting days limited the intake to 300–500 kcal to support targeted BMI reduction. The primary goals of dietary modification were to detoxify and alkalize the body and gradually reduce calorie intake through intermittent fasting and juice fasting to promote weight loss [13,14]. After achieving their initial BMI goals, participants transitioned to a 1400–1600 kcal/day diet with 1–1.2 grams of protein per kilogram of body weight to support muscle gain alongside exercise. The final phase focused on long-term glycemic control, increasing caloric intake to 1600–1800 kcal/day, with additional protein as per exercise needs, under the supervision of a dietitian and a physical therapist. This stage aimed to help participants sustain weight loss and maintain their dietary and exercise routines.

The diet plan was complemented with exercises to improve strength, flexibility, and stamina. Phase 1 emphasized improving lymphatic circulation, muscle activation, and anti-gravity exercises, including warm-ups, *Sooryanamaskaras* (sun salutations), super brain yoga [15] and palm planks, for post-meal glycemic control. Phase 2 emphasized on muscle gain, weight loss, and core strengthening. Phase 3 introduced personalized activities, such as swimming, running, cycling, or yoga, based on comorbidities, age, and preferences for athletic specializations. In the final phase, periodized plans were implemented to sustain strength, stamina, and flexibility.

Psychological support was aimed at stress and anxiety management and enabled mind-body awareness. This intervention comprised of online group therapy sessions and individual counseling sessions by trained psychologists (upon request by patients) using specific therapeutic approaches, including cognitive behavior therapy (CBT), rational emotional behavior therapy (REBT), neuro-linguistic programming (NLP), clinical hypnotherapy, life coaching, and *pranic* healing [16].

Finally, medical management included physician-directed daily medication adjustments via a dedicated mobile application based on self-reported glucose readings, correction of micronutrient deficiencies through supplementation, quarterly physician consultations, and ongoing care for pre-existing comorbidities. Medication tapering was performed in a stepwise manner, tailored to each participant's clinical progress, and supported by daily BGL submitted via a mobile application. Treating physicians closely monitored BGL trends to make timely dose modifications. The deprescription process was informed by the RSSDI–ESI Therapeutic Wheel, prioritizing the withdrawal of sulfonylureas and insulin while continuing metformin or DPP-4 inhibitors when clinically appropriate [17].

**Mode of delivery.** A dedicated smartphone application enabled patients to interact with their expert team through voice calls, video calls, and text messages. Clinical data, including self-monitored blood glucose entries from the mobile application and electronic medical records, were linked using a unique participant ID generated at enrollment. The mobile app was integrated with a central data system to enable secure real-time synchronization. At the time of enrollment in the intervention program, participants digitally consented to the use of their anonymized data for research and quality improvement purposes as part of routine onboarding.

During the intervention period, patients were required to regularly report vital signs (blood glucose and weight) through the application, allowing clinicians to adjust medications as needed and guide the diet and exercise intervention. The application also provided region-specific plant-based recipes and pre-recorded exercise and meditation audios, supporting personalized diet, exercise, and stress management plans.

Adherence was supported through a structured care model that combined daily self-monitoring of BGL with continuous engagement from the care team. Participants received weekly or fortnightly consultations with dietitians and physical activity coaches, which later transitioned to monthly and quarterly physician reviews. A mobile application enabled real-time tracking and communication. On average, participants received 63 calls annually and attended monthly online group sessions focused on education and motivation, helping sustain adherence throughout the intervention.

Ethical approval for this study was obtained from the Institutional Ethics Committee (FFDRF/IEC/2024/7), which also waived the requirement for additional written informed consent in accordance with the national guidelines for retrospective research involving minimal risk. This study is registered with the Clinical Trials Registry of India (CTRI/2024/03/064596). The authors had no access to personally identifiable information, thereby ensuring participant confidentiality. This study was conducted in accordance with the Declaration of Helsinki and its amendments.

## Statistical analyses

Categorical variables are presented as frequencies and percentages and continuous variables as medians with interquartile ranges owing to the skewed data distribution. The % change was calculated as the final value minus the initial value divided by the initial value multiplied by 100 to adjust for baseline differences. Pre- and post-intervention comparisons, both for the total cohort and subgroup analyses, were conducted using a linear mixed-effects model (LMM) to account for repeated measurements and adjust for relevant covariates. To account for multiple comparisons across outcome variables, we applied the Benjamini–Hochberg (BH) procedure to control the false discovery rate at 5%. BH-adjusted p-values were computed separately for each set of comparisons (Time, Group, and Group×Time effects) in the LMMs. Adjusted p-values <0.05 were considered statistically significant [18].

The chi-square test was used to examine the associations between categorical variables. Variables showing significant bivariate associations with remission were included in the multivariable logistic regression analysis to identify independent predictors of diabetes remission (yes/no): age, HbA1c, medication status, weight change, HOMA-IR, juice fasting, intermittent fasting status, and disease duration. Continuous variables were categorized using clinically relevant thresholds (e.g., HbA1c <7% [19], weight loss >10% [20], and HOMA-IR ≥2.5 [21]) to enhance interpretability. For diabetes duration, a 6-year cut-off was applied based on existing evidence suggesting higher remission rates in patients with shorter disease duration [22]. This threshold enabled meaningful subgroup comparisons, despite the

cohort's median duration of 7.6 years. Statistical analyses were performed using the IBM SPSS ver. 21, with statistical significance set at $p < 0.05$.

## Patient and public involvement

Patients and/or the public were not involved in the design, conduct, reporting, or dissemination of this research.

## Results and discussion

### Baseline characteristics of the study population

Sociodemographic and anthropometric parameters along with the medication status of the population (n = 2384) at the start of the intervention (baseline) are shown in Table 1. The median age of the patients was 51 years (interquartile range [IQR]: 44.0, 58.0) years with 60.1% males. The median diabetes duration was 7.6 (IQR: 3.6, 12.3) years. Median BMI, HbA1c levels, FBG, fasting insulin, HOMA-IR, and HOMA-β were 27.0 (IQR: 24.9, 29.9) kg/m$^2$, 7.5 (IQR: 6.7, 8.7) %, 130.6 (IQR: 111.0, 157.0) mg/dl, 9.5 (IQR: 6.4, 14.4) μU/ml, 3.2 (IQR: 2.0, 5.1), and 51.1 (IQR: 29.3,87.6), respectively. Poor glycemic control (HbA1c ≥7%) was observed in the majority of the population (67.3%), despite 88.1% being on OHAs. The median HbA1c levels in drug-naïve and OHA-treated patients were 6.9 (IQR: 6.5, 7.5) % and 7.6 (IQR: 6.8, 8.9) %, respectively.

### Post-intervention improvements in anthropometric and biochemical parameters based on remission status

Post-intervention, 31.2% (n = 744) of the participants achieved diabetes remission. Among drug-naïve participants, HbA1c improved significantly after the intervention [post-intervention: 6.0% (IQR: 5.6–6.3) vs. baseline: 6.9% (IQR: 6.5–7.5); $p < 0.05$], highlighting the effectiveness of lifestyle modification alone in improving glycemic control.

Table 2 provides a detailed comparison of anthropometric and biochemical parameters between the remission and non-remission groups at both the baseline and post-intervention, along with estimates from the LMM analysis. From baseline to post-intervention, the intervention resulted in significant improvements in weight, BMI, HbA1c, FBG, fasting insulin, and HOMA-IR across the entire cohort; no significant change was observed for HOMA-β.

Groupwise, significant differences were observed between the remission and non-remission group, where at both time points, participants in the remission group consistently had lower weight, BMI, HbA1c, FBG, fasting insulin, HOMA-IR, and higher HOMA-β values (all $p < 0.05$), indicating better glycemic and metabolic status.

Interaction analysis (Group x time) showed that participants in the remission group experienced a significantly greater reduction in weight, BMI, HbA1c, fasting insulin, and HOMA-IR (all $p < 0.05$). The changes in FBG and HOMA-β did not differ significantly between the groups ($p > 0.1$). However, the relatively lower FBG levels and higher HOMA-β in the remission group suggested better glycemic control and preservation of β-cell function compared to the non-remission group.

Among the adjusted covariates, younger age was significantly associated with higher weight, BMI, FBG, and HOMA-IR (all $p < 0.05$). Regardless of time and remission status, females had lower weight and higher BMI than males ($p < 0.05$), likely due to the height difference in the two sexes. Additionally, longer diabetes duration was associated with higher HbA1c and FBG levels and lower fasting insulin, HOMA-IR, and HOMA-β ($p < 0.05$).

### Sustainability of glycemic control and remission Post-OHAs discontinuation

At one-year follow-up, 41.4% of participants had discontinued all OHAs and remained off-medication for a median duration of 9 months (IQR: 5.0–11.3). Overall, 744 participants (31.2%) achieved diabetes remission, defined as maintaining HbA1c < 6.5% without OHAs for at least three months.

Among drug-naïve participants (n = 283; 11.9% of the cohort), the overall remission rate was 83%. Of the total, 22.9% (n = 65) had baseline HbA1c < 6.5%, and 21.9% (n = 62) maintained this level post-intervention. Excluding these individuals, the true remission rate, defined as a reduction from HbA1c ≥6.5% (n = 218) to <6.5% without medications, was 79.3%

**Table 1. Baseline characteristics of the study population.**

| Parameter | | N (%) |
|---|---|---|
| *Anthropometric, Socio-demographic, and Clinical Characteristics; n = 2384* | | |
| Age (years) | ≤50 | 1175 (49.3) |
| | >50 | 1209 (50.7) |
| Sex | Male | 1432 (60.1) |
| | Female | 952 (39.9) |
| Marital status | Married | 2188 (91.8) |
| | Unmarried | 67 (2.8) |
| | Widowed/Divorced | 129 (5.4) |
| Education | Up to 12 years of schooling | 105 (4.4) |
| | Graduate | 1005 (42.2) |
| | Post-graduate and above | 1191 (50.0) |
| | Others* | 83 (3.5) |
| Occupation | Employed | 1633 (68.5) |
| | Homemaker | 425(17.8) |
| | Retired | 227 (9.5) |
| | Others* | 99 (4.2) |
| Family history of diabetes | Yes | 1877 (78.7) |
| | No | 507 (21.3) |
| BMI classification | Overweight (BMI 23.0–24.9 kg/m²) | 604 (25.3) |
| | Obese (BMI ≥ 25.0 kg/m²) | 1780 (74.7) |
| *Glycemic Parameters and Medication Status n = 2384* | | |
| HbA1c (%) | <7.0 | 779 (32.7) |
| | ≥7.0 | 1605 (67.3) |
| Fasting Blood Glucose (mg/dl) | <100 | 288 (12.1) |
| | ≥100 | 2096 (87.9) |
| Fasting Insulin (µU/ml); n = 1897 [23] | <25 | 1795 (94.6) |
| | ≥25 | 102 (5.4) |
| HOMA-IR; n = 1897 [21] | <2.5 | 669 (35.3) |
| | ≥2.5 | 1228 (64.7) |
| Medication status | OHAs | 2101 (88.1) |
| | Drug naive | 283 (11.9) |

Data are presented as frequency and %; BMI, body mass index; OHAs, oral hypoglycemic agents; HbA1c, glycated hemoglobin; HOMA-IR, homeostatic model assessment of insulin resistance; [21,23] references for corresponding cutoff values of the parameters. Fasting Insulin and HOMA-IR data were reported for a subset (n = 1897) for which complete data (pre- and post-intervention) for fasting insulin were available. *Did not share the details.

(n = 173). These individuals sustained HbA1c <6.5% for a median duration of 8.6 months (IQR: 7.6–9.5), while 20.6% (n = 45) remained above the threshold of 6.5%. In comparison, participants on OHAs had a significantly lower remission rate of 24.2%.

## Remission and its association with parameters

To further explore factors associated with remission, bivariate analysis was performed (Fig 2a, 2b). Remission was more likely in participants aged ≤50 years, with BMI ≥ 25 kg/m², baseline HbA1c <7%, disease duration ≤6 years, and drug-naïve

**Table 2. Comparison of clinical and biochemical parameters at baseline and post-intervention based on remission status.**

| Outcome | Group | Baseline | Post-intervention | % Change (95% CI) | Model Effect | Estimate (95% CI) | BH adjusted p value¥ |
|---|---|---|---|---|---|---|---|
| Weight (kg) | Total cohort | 75.0 (68.0, 84.0) | 70.3 (63.4, 78.5) | −6.0 (−6.3, −5.8) | Time (Baseline vs post-intervention) | 4.12 (3.90, 4.35) | <0.001 |
| | Remission | 76.0 (69.0, 85.0) | 70.0 (63.0, 77.2) | −8.5 (−9.1, −8.1) | Group (Remission vs non-remission) | −1.55 (−2.59, −0.51) | 0.005 |
| | Non-Remission | 74.3 (67.7, 84.0) | 70.5 (64.0, 79.0) | −5.2 (−5.6, −4.9) | Group * Time | 2.77 (2.37, 3.17) | <0.001 |
| | | | | | Age | −0.15 (−0.20, −0.10) | <0.001 |
| | | | | | Sex (Female/Male) | −7.20 (−8.19, −6.22) | <0.001 |
| BMI (kg/m²) | Total cohort | 27.0 (24.9, 29.9) | 25.2 (23.5, 27.9) | −6.0 (−6.3, −5.9) | Time (Baseline vs post-intervention) | 1.53 (1.44, 1.61) | <0.001 |
| | Remission | 27.4 (25.3, 30.7) | 24.9 (23.3, 27.7) | −8.6 (−9.3, −8.3) | Group (Remission vs non-remission) | −0.36 (−0.70, 0.03) | 0.031 |
| | Non-Remission | 26.8 (24.0, 29.0) | 25.3 (23.5, 28.0) | −5.2 (−5.6, −5.0) | Group * Time | 1.01 (0.86, 1.17) | <0.001 |
| | | | | | Age | −0.04 (−0.06, −0.02) | <0.001 |
| | | | | | Sex (Female/Male) | 1.69 (1.37, 2.00) | <0.001 |
| HbA1c (%) | Total cohort | 7.5 (6.7, 8.7) | 6.4 (5.9, 7.0) | −13.5 (−14.1, −12.8) | Time (Baseline vs post-intervention) | 1.30 (1.21, 1.38) | <0.001 |
| | Remission | 6.9 (6.4, 7.8) | 5.9 (5.6, 6.1) | −15.3 (−16.4, −14.3) | Group (Remission vs non-remission) | −1.00 (−1.07, −0.93) | <0.001 |
| | Non-Remission | 7.9 (7.0, 9.1) | 6.8 (6.3, 7.3) | −12.4 (−13.3, −11.6) | Group * Time | 0.19 (0.05, 0.34) | 0.008 |
| | | | | | Age | −0.007 (−0.110, −0.004) | <0.001 |
| | | | | | Diabetes duration | 0.02 (0.015, 0.026) | <0.001 |
| FBG (mg/dl) | Total cohort | 130.6 (111.0,157.0) | 116.3 (101.7,134.0) | −10.3 (−11.6, −9.3) | Time (Baseline vs post-intervention) | 19.76 (17.58, 21.93) | <0.001 |
| | Remission | 119.0 (104.7,136.9) | 103.0 (94.0, 114.6) | −13.2 (−14.5, −11.2) | Group (Remission vs non-remission) | −22.46 (−24.91, −20.01) | <0.001 |
| | Non-Remission | 138.0 (116.0, 165.9) | 124.1 (108.6, 143.0) | −8.8 (−10.3, −7.4) | Group * Time | 0.87 (−3.06, 4.79) | 0.777 |
| | | | | | Age | −0.40 (−0.51, −0.28) | <0.001 |
| | | | | | Diabetes duration | 0.58 (0.39, 0.77) | <0.001 |
| Fasting Insulin (µU/ml)ᵃ | Total cohort | 9.5 (6.4,14.4) | 7.9 (5.1,11.6) | −15.9 (−18.3, −13.2) | Time (Baseline vs post-intervention) | 1.81 (1.42, 2.20) | <0.001 |
| | Remission | 10.5 (7.1, 15.6) | 7.3 (5.0, 11.7) | −26.6 (−30.5, −23.1) | Group (Remission vs non-remission) | −0.94 (−1.64, −0.24) | 0.009 |
| | Non-Remission | 9.2 (6.1,14.0) | 8.0 (5.1, 11.6) | −11.4 (−14.3, −8.9) | Group * Time | 1.48 (0.77, 2.20) | <0.001 |
| | | | | | Age | −0.02 (−0.050, 0.008) | 0.153 |
| | | | | | Diabetes duration | −0.16 (−0.21, −0.11) | <0.001 |
| HOMA-IRᵃ | Total cohort | 3.2 (2.0, 5.1) | 2.3 (1.4,3.5) | −24.3 (−27.7, −21.5) | Time (Baseline vs post-intervention) | 1.09 (0.93, 1.25) | <0.001 |
| | Remission | 3.1 (2.0, 5.0) | 1.9 (1.2, 3.0) | −37.3 (−41.2, −32.8) | Group (Remission vs non-remission) | −0.84 (−1.09, −0.59) | <0.001 |
| | Non-Remission | 3.2 (2.0, 5.2) | 2.4 (1.5, 3.7) | −19.7 (−22.8, −16.3) | Group * Time | 0.40 (0.09, 0.70) | 0.013 |
| | | | | | Age | −0.02 (−0.03, −0.01) | 0.002 |
| | | | | | Diabetes duration | −0.04 (−0.05, −0.02) | <0.001 |

*(Continued)*

**Table 2.** (Continued)

| Outcome | Group | Baseline | Post-intervention | % Change (95% CI) | Model Effect | Estimate (95% CI) | BH adjusted p value¥ |
|---------|-------|----------|-------------------|-------------------|--------------|-------------------|----------------------|
| HOMA-β[a] | Total cohort | 51.1 (29.3,87.6) | 52.8 (32.4,85.2) | 3.3 (0.05, 7.30) | Time (Baseline vs post-intervention) | −12.37 (−36.61, 11.86) | 0.317 |
| | Remission | 71.0 (42.9, 113.2) | 71.3 (44.9, 112.1) | 2.9 (−1.2, 11.4) | Group (Remission vs non-remission) | 23.89 (16.50, 31.29) | <0.001 |
| | Non-Remission | 44.9 (26.5, 75.1) | 47.3 (29.5, 75.8) | 3.4 (−1.0, 8.0) | Group * Time | 8.57 (−36.98, 54.12) | 0.710 |
| | | | | | Age | 0.30 (−0.05, 0.66) | 0.103 |
| | | | | | Diabetes duration | −1.37 (−1.96, −0.79) | <0.001 |

Data are presented as the median (IQR), % change is presented as median and 95% CI and reflects unadjusted change from baseline; [a] number of patients (n) for Fasting Insulin, HOMA-IR, and HOMA-β −1897 (total), 545 (remission), 1352 (non-remission); Linear Mixed Model (LMM) analyses were performed to estimate adjusted mean differences and interactions using: HbA1c outcome adjusted for age and diabetes duration, Weight and BMI outcomes adjusted for age and sex, Fasting blood glucose, HOMA-IR, and HOMA-β outcomes adjusted for age and diabetes duration; ¥ Adjusted p values were calculated using the Benjamini-Hochberg (BH) procedure to control the false discovery rate (FDR) at α = 0.05. The BH critical threshold for each hypothesis (ranked by p value) was computed as (i/m) × α, where i = rank and m = total number of tests. Adjusted p values <0.05 were considered statistically significant. Reference category: group – non-remission, sex male; BMI, body Mass Index; HOMA-IR, homeostatic model assessment of insulin resistance; HbA1c, glycated hemoglobin; HOMA-β, Homeostatic Model Assessment of β-cell function; FBG, Fasting blood glucose.

status (Fig 2a). Other post-intervention factors associated with remission were adherence to juice and intermittent fasting, HbA1c improvement of >15%, weight loss of >10%, and reduction in insulin resistance (HOMA-IR < 2.5) (p<0.05) (Fig 2b). No sex-wise association was observed with remission (p>0.1).

Post-intervention, the improvement in HbA1c (>15% decrease) was similar between drug-naïve patients (45.9%) and those on OHAs (45.4%) (p>0.1), indicating the efficacy of the intervention irrespective of the initial medication status. However, post-intervention weight loss (>10%) was significantly higher in drug-naïve patients (31.8%) than in those receiving OHAs (24.2%); (p = 0.005).

Although remission was significantly associated with weight loss, 58 patients (22.8%) who gained weight post-intervention (n = 254) also achieved remission. The median weight gain in these patients was 1.5 (IQR; 1.0, 3.0) kg. The median diabetes duration and HbA1c level at baseline were 3.7 (IQR: 2.9, 7.2) years and 6.6 (IQR: 6.0, 7.3) %, respectively. Thirty-one percent of these participants were drug naïve.

We further explored the relationship between remission status, decreased HbA1c level, and weight loss (S1A, S1B, S1C Fig). Greater weight loss and decreased HbA1c levels were associated with an increased remission rate (S1A, S1B Fig). A significant positive association between weight loss and HbA1c levels (p <0.05) was also observed (S1C Fig).

## Predictors of remission

Binomial logistic regression was performed to assess predictors of remission. The analysis was done on a subset of patients (n = 1897) with the complete data (pre-post-intervention) for fasting insulin, used to calculate HOMA-IR. A shorter disease duration, younger age, drug-naïve status, good HbA1c control at baseline, juice fasting, >10% weight loss, and lower insulin resistance (HOMA-IR) post-intervention were significant predictors of remission (p<0.05) (Fig 3).

## Discussion

The increasing prevalence of T2D and its associated comorbidities requires effective alternatives to conventional pharmacotherapy. Multidisciplinary lifestyle interventions have shown promise for T2D management and remission. This study reports the characteristics and population-specific predictors of remission in a large Indian cohort. Unlike previous intervention studies, continued medical management was part of the program, and no drastic dietary changes were introduced

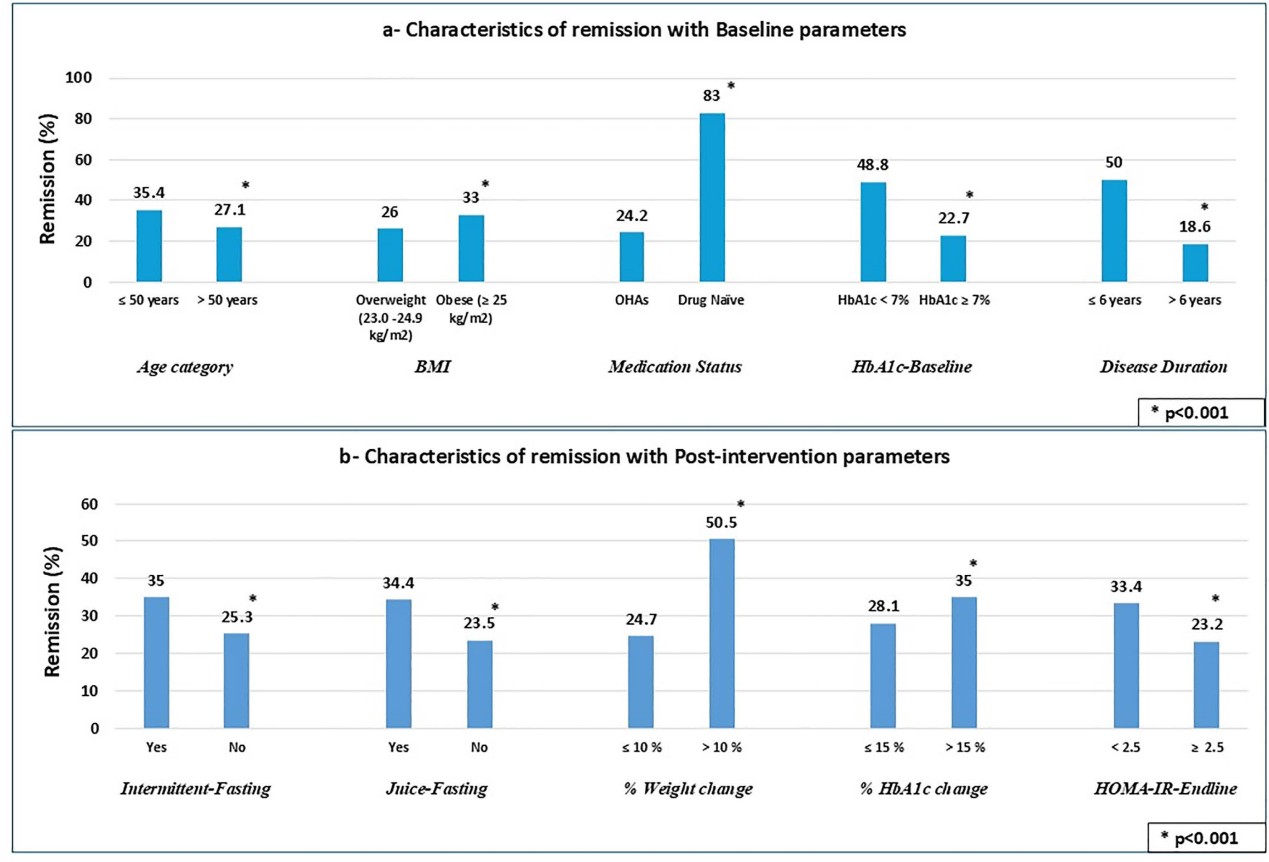

**Fig 2. Characteristics of remission. [a]** Baseline and **[b]** post-intervention parameters. OHAs, oral hypoglycemic agents; HbA1c, glycated hemoglobin; HOMA-IR, homeostatic model assessment of insulin resistance; For HOMA-IR, analysis was done on a subset of patients n = 1897 with the complete data (pre-post-intervention) for fasting insulin, used to calculate HOMA-IR.

to achieve remission. Considering the phenotypic and genotypic diversity of the Indian population compared with Western cohorts, these findings provide valuable evidence supporting non-pharmacological diabetes interventions [4,5,9].

We observed a remission rate of 31.2%, highlighting the potential of lifestyle interventions to achieve diabetes remission. This remission rate aligns with a recent study reporting a 20% remission rate through one-to-one consultations emphasizing weight loss and low-carbohydrate diets [24]. In contrast, higher remission rates have been reported in other randomized controlled trials with smaller cohorts [25,26]. Nevertheless, these findings emphasize the significance of a multidisciplinary approach to T2D remission.

Initial medication status played a significant role, with individuals who had never started medication demonstrating a higher likelihood of remission, thus highlighting the importance of early lifestyle intervention before opting for pharmacotherapy. Additionally, weight loss emerged as the strongest predictor of remission, reinforcing its role in T2D management. This is consistent with other studies that have highlighted diet and weight loss as central strategies for diabetes remission [4,9]. Our study indicated that individuals with a weight loss >10% had a significantly higher remission rate.

Another interesting finding was that some patients achieved remission despite post-intervention weight gain, although this percentage was low. This contrasts with the Personal Fat Threshold theory proposed by Taylor et al., where remission

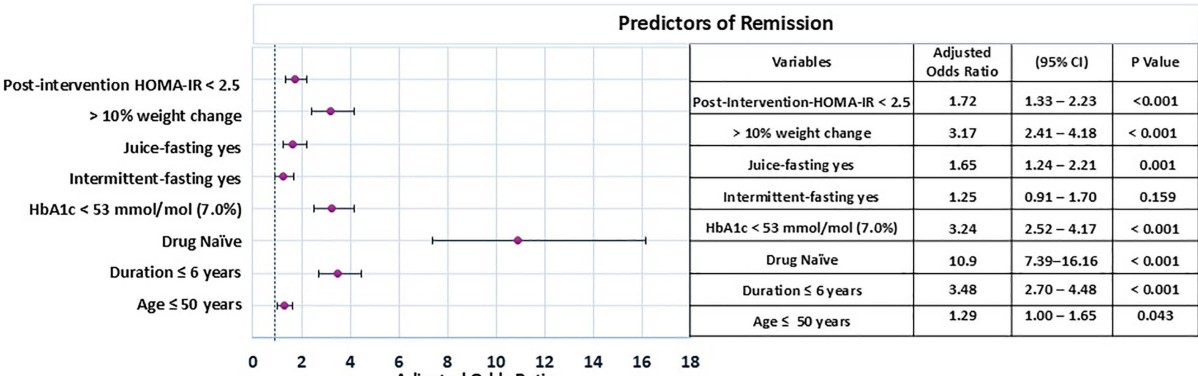

**Fig 3. Predictors of T2D Remission (n = 1897).** *Footnote:* The forest plot displays adjusted odds ratios (AORs) with 95% confidence intervals (CI) for various parameters to identify predictors. Each circle represents the point estimate, and the horizontal lines show the 95% confidence intervals, which reflect the range within which the true AOR falls. The dashed vertical line at AOR = 1 represents the line of no effect for predictors post-intervention.

is linked to weight loss in individuals with low BMI [27]. However, the weight gain was minimal and these patients had a mean baseline HbA1c of 6.6%, shorter diabetes duration, and >30% were drug-naïve, possibly contributing to remission. This may also be due to initial weight loss during the intervention. Whether this weight gain is due to muscle building or fat accumulation requires further investigation. Furthermore, a long-term follow-up is required to determine whether remission was sustained in these patients.

Baseline factors such as age (≤50 years), diabetes duration (≤6 years), and baseline HbA1c level (<7.0%) predicted remission, regardless of the initial BMI, highlighting the importance of early intervention. Furthermore, post-intervention improvements in HbA1c and insulin resistance (HOMA-IR) were associated with remission. Our findings, except for younger age (≤50 years), align with the Diabetes Remission Clinical Trial (DiRECT), especially regarding medication use, baseline HbA1c, and weight loss [4,28]. A significant association between reduced HOMA-IR following weight loss and remission was observed, consistent with studies linking low insulin resistance with sustained remission in overweight and obese patients undergoing calorie restriction [29–31]. This emphasizes the need for effective weight management and dietary interventions that specifically address insulin resistance. Intermittent fasting and juice fasting were associated with remission, with juice fasting emerging as a novel predictor. Intermittent fasting has gained attention for its potential impact on metabolic health, and its positive association with remission is consistent with previous research [31]. This finding suggests that incorporating fasting periods into diabetes management may improve T2D outcomes. While some plant-based juices may have potential benefits in cardiovascular diseases because of their nutrient and detoxifying properties, studies suggest that juice fasting may not provide substantial benefits for T2D management, which contradicts our findings [32,33]. Further research is needed to determine the role of juice fasting in glycemic control and overall diabetes management.

In our study, no significant changes in HOMA-β values were observed post-intervention. However, when the baseline HOMA-β values were compared between the remission and non-remission groups, a significant difference was observed in the remission group, with a higher baseline HOMA-β. A higher HOMA-β value at baseline suggests a greater capacity of β-cells to produce insulin, which may have been a key factor in achieving remission in these individuals. Similar observations have been reported in previous studies, in which participants who achieved diabetes remission had higher baseline β-cell function [34,35]. There was no association between sex and remission, indicating that both males and females had an equal chance of achieving remission.

Our study, despite its valuable findings, has several limitations. The retrospective nature of the research and the lack of a control group limit our ability to attribute observed outcomes solely to the intervention. Additionally, the one-year program was a subscription-based service, which may have restricted access to those who could afford it, skewing the participant pool towards individuals with higher socioeconomic status. This group is typically associated with better health outcomes regardless of intervention, which may have impacted the generalizability of the results. Moreover, participants' higher socioeconomic and educational backgrounds might have influenced their ability to adhere to the program, possibly affecting remission rates. We also acknowledge that the exclusion of participants with limited follow-up and missing data may have introduced selection bias. While objective adherence measures were not recorded, adherence was supported by regular consultations, daily glucose tracking, and interactive digital engagement. Although we had a large sample size, the retrospective design may have introduced selection bias and confounding factors, potentially limiting the broader applicability of our findings to the Indian population. Further, the medication tapering followed a structured principle-based clinical protocol (RSSDI–ESI Therapeutic Wheel) and was guided by real-time glycemic trends, yet the retrospective nature of the study may have allowed for some variability in its implementation. Differences in physician judgment, documentation practices, and individual patient responses may have influenced the consistency of OHA withdrawal among participants. While this approach reflects real-world clinical practice, prospective studies with uniform deprescription algorithms and predefined remission criteria are warranted to validate these findings.

On the positive side, the online nature of the program allowed us to reach patients from 263 cities across India, broadening the scope and diversity of our sample. However, this also meant that participants were likely to possess a certain level of digital literacy, which could affect adherence and, consequently, the effectiveness of the intervention. Despite these limitations, our research offers meaningful insights into the real-world effectiveness of intensive lifestyle intervention strategies. Future prospective studies will be crucial in further corroborating these findings.

## Conclusions

Our findings add to the growing body of evidence supporting lifestyle interventions as effective strategies for achieving T2D remission in the Indian population. Absolute changes in weight, juice fasting, and post-intervention HOMA-IR were identified as novel remission predictors specific to the Indian population. These results emphasize the importance of personalized interventions targeting multiple aspects of T2D management to improve remission rates and long-term health outcomes. Long-term follow-up and planned randomized controlled trials may help understand the sustainability and feasibility of remission achieved through ILI.

## Supporting information

**S1 Fig. Associations between remission, HbA1c reduction, and weight loss.**
(PDF)

**S2 File. Dataset.**
(XLSX)

## Acknowledgments

The authors wish to thank the entire team at Freedom from Diabetes Clinic. Additionally, we wish to acknowledge the use of the Paperpal software (www.paperpal.com) for copy-editing the language. Following the use of this tool, all authors examined and modified the content as necessary, assuming full accountability for the publication material.

## Author contributions

**Conceptualization:** Pramod Tripathi, Nidhi Kadam.

**Data curation:** Nidhi Kadam, Thejas Kathrikolly, Baby Sharma.

**Formal analysis:** Nidhi Kadam, Diptika Tiwari, Anagha Vyawahare.

**Investigation:** Thejas Kathrikolly.

**Methodology:** Nidhi Kadam, Thejas Kathrikolly, Diptika Tiwari, Anagha Vyawahare, Baby Sharma.

**Project administration:** Nidhi Kadam.

**Supervision:** Pramod Tripathi, Malhar Ganla, Banshi Saboo.

**Validation:** Pramod Tripathi.

**Writing – original draft:** Nidhi Kadam, Thejas Kathrikolly.

**Writing – review & editing:** Pramod Tripathi, Nidhi Kadam, Thejas Kathrikolly, Diptika Tiwari, Anagha Vyawahare, Baby Sharma, Malhar Ganla, Banshi Saboo.

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
