## [Decision Letter · Decision Letter 0]

2 Jul 2025

Dear Dr. Kadam,

Thank you for submitting your manuscript to PLOS ONE. After careful consideration, we feel that it has merit but does not fully meet PLOS ONE’s publication criteria as it currently stands. Therefore, we invite you to submit a revised version of the manuscript that addresses the points raised during the review process.

**Please address reviewers' comments.**plosone@plos.org . A rebuttal letter that responds to each point raised by the academic editor and reviewer(s). You should upload this letter as a separate file labeled 'Response to Reviewers'.A marked-up copy of your manuscript that highlights changes made to the original version. You should upload this as a separate file labeled 'Revised Manuscript with Track Changes'.An unmarked version of your revised paper without tracked changes. You should upload this as a separate file labeled 'Manuscript'.

We look forward to receiving your revised manuscript.

Kind regards,

Guoying Wang, MD, PhD

Academic Editor

PLOS ONE

**Journal Requirements:**

1. When submitting your revision, we need you to address these additional requirements. Please ensure that your manuscript meets PLOS ONE's style requirements, including those for file naming. The PLOS ONE style templates can be found at https://journals.plos.org/plosone/s/file?id=wjVg/PLOSOne_formatting_sample_main_body.pdf and https://journals.plos.org/plosone/s/file?id=ba62/PLOSOne_formatting_sample_title_authors_affiliations.pdf 2. Your ethics statement should only appear in the Methods section of your manuscript. If your ethics statement is written in any section besides the Methods, please delete it from any other section. 3. Please include captions for your Supporting Information files at the end of your manuscript, and update any in-text citations to match accordingly. Please see our Supporting Information guidelines for more information: http://journals.plos.org/plosone/s/supporting-information.

Reviewers' comments:

Reviewer's Responses to Questions

**Comments to the Author**

1. Is the manuscript technically sound, and do the data support the conclusions?

Reviewer #1: Yes

Reviewer #2: Partly

2. Has the statistical analysis been performed appropriately and rigorously?

Reviewer #1: Yes

Reviewer #2: No

3. Have the authors made all data underlying the findings in their manuscript fully available?

Reviewer #1: Yes

Reviewer #2: No

4. Is the manuscript presented in an intelligible fashion and written in standard English?

Reviewer #1: Yes

Reviewer #2: Yes

**Reviewer #1:**  In this paper, the authors reported predictors of type 2 diabetes remission in an Indian cohort, by using their unique intervention program based on Indian culture. The reported predictors per se are not so novel, in agreement with the previous papers, but unique in that the authors employed their original program. The reviewer’s criticisms are as follows.

Major comments

1. Diabetes remission (lines 121-122) is defined as achieving HbA1c levels <6.5% without the use of oral hypoglycemic agents (OHA) for at least 3 months. At baseline, ~90% of the participants are treated with OHAs (Table 1). Are there any guidelines to stop OHAs in this study? Otherwise, OHAs are stopped only by physicians’ discretion? This point should be clarified.

2. Regarding above point, if there are no guidelines to stop OHAs, and also this is a retrospective study, the limitation section should be enriched. Diabetes remission seems to be highly arbitrary.

3. Results, Fig. 2a and Fig. 3. Diabetes duration is analyzed by the cut-off of 6 years, while median diabetes duration is 7.6 years (line 202). Why the cut-off of 6 years is adopted?

Minor comments

1. Eligibility criteria, line 100. Peripheral artery disease is not a part of neuropathy. Revise appropriately.

2. Table 1, line 213. Ref [15. 16] may be [16, 17]. Check the accuracy.

**Reviewer #2: ** This study seeks to evaluate the remission of T2D in an Indian cohort using a culturally sensitive 4-component lifestyle intervention. Several concerns and questions are noted:

The statistical analysis section is poorly developed.

• For the univariate logistic regression, the authors choose to convert continuous variables into categorical ones without any justification. The categorical cut-offs appear arbitrary, and the use of categories where continuous does not take full advantage of all the data available in the study. Moreover, why weren’t all the other variables in table 2 included in the logistic regression (e.g. weight, FBS)?

• The generation and use of p-values in table 2 is confusing and distracting. Why are two different statistical tests being used for data with the same properties? Additionally, the authors reused the same data multiple times, but did not account for this in their analysis (i.e. no multiple comparison corrections).

• A multivariable logistic regression analysis should be performed.

This authors state that this study was conducted at a Freedom from Diabetes Clinic using a 1 year online subscription-based model. It appears that the term post-intervention refers to the one year follow up period. There is no data presented that documents the length of time a patient was off OHAs at the one year mark nor how long drug naïve patients were found to be <6.5%. This should be included, as it is a central part of the study that is used to define remission.

All participants had to pay for the use of this subscription, and the resulting limitations are appropriately acknowledged and discussed. However, the exclusion of participants due to missing or incomplete data also introduces bias. Adherence and non-compliance should be reported, considered statistically, and discussed. How many people were not able to complete this program and why?

There is no description of how self-reported data from a phone app was matched to the electronic medical records of a patient, if at all, and whether the patient consented to use of such information. The authors should elaborate on or provide additional documentation to substantiate the following statement: “The requirement for informed consent was waived by the Ethics Committee owing to the retrospective nature of the study”

It appears HbA1c values were taken from the electronic medical record, but this was not made clear. Since data is being obtained online from participants in geographically diverse Indian cities (i.e. 263 Indian cities), how do the authors know that the blood samples taken were sent to laboratories that have standardized HbA1c assays that align with international reference standards?

**Do you want your identity to be public for this peer review?** For information about this choice, including consent withdrawal, please see our Privacy Policy

Reviewer #1: No

Reviewer #2: No

---

## [Author Response · Author response to Decision Letter 1]

7 Aug 2025

RESPONSE TO REVIEWERS

We sincerely thank the editor and the reviewers for their valuable and insightful comments; they helped improve the quality of our manuscript. Please find below the detailed response to the reviewers’ comments for your kind perusal and necessary consideration.

Reviewer Comments and Response

Reviewer 1

In this paper, the authors reported predictors of type 2 diabetes remission in an Indian cohort, by using their unique intervention program based on Indian culture. The reported predictors per se are not so novel, in agreement with the previous papers, but unique in that the authors employed their original program.

The reviewer’s criticisms are as follows.

Major comments

1. Diabetes remission (lines 121-122) is defined as achieving HbA1c levels <6.5% without the use of oral hypoglycemic agents (OHA) for at least 3 months. At baseline, ~90% of the participants are treated with OHAs (Table 1). Are there any guidelines to stop OHAs in this study? Otherwise, OHAs are stopped only by physicians’ discretion? This point should be clarified

Response: Thank you for your comment. We appreciate the reviewer highlighting this important point. In our study, medication stoppage was not initiated at baseline (as in the DiRECT trial) but was progressively implemented based on individual clinical progress assessed through routine follow-up. A key component of our strategy was the use of self-reported daily blood glucose level (BGL) monitoring through a mobile application, enabling patients to submit their blood glucose levels in real time. Treating physicians reviewed these data daily and made day-to-day drug dose adjustments, allowing for responsive deprescription while ensuring glycemic stability and minimizing hypoglycemia risk. A structured deprescription approach was followed which was guided by the RSSDI-ESI Therapeutic Wheel (https://doi.org/10.1007/s13410-020-00819-2), a tool developed specifically for the Indian population. This framework considers HbA1c levels, diabetes duration, weight loss, hypoglycemia risk, and comorbidities to individualize decisions. The deprescription process was not rigidly protocolized but followed a principle-based algorithm:

• Patients experiencing stable or normal BSL trends as a result of the intervention, without hypoglycemia episodes, were considered for stepwise medication tapering.

• Tapering was guided by daily BSL monitoring through a mobile application, enabling real-time tracking and physician-supervised day-to-day medication adjustment.

• Sulfonylureas and insulin were prioritized for withdrawal, particularly when hypoglycemia risk was evident.

• Metformin and DPP-4 inhibitors were generally retained longer due to their favorable profiles.

All deprescription decisions were made by the treating physician using clinical judgment, supported by the above framework. This individualized yet structured approach enabled safe and effective deprescription.

We have now clarified this methodology in the Materials and Methods section as follows- (Page no.7, Lines 184-189).

“Medication tapering was performed in a stepwise manner, tailored to each participant’s clinical progress, and supported by daily BGL submitted via a mobile application. Treating physicians closely monitored BGL trends to make timely dose modifications. The deprescription process was informed by the RSSDI–ESI Therapeutic Wheel, prioritizing the withdrawal of sulfonylureas and insulin while continuing metformin or DPP-4 inhibitors when clinically appropriate [17].”

2. Regarding above point, if there are no guidelines to stop OHAs, and also this is a retrospective study, the limitation section should be enriched. Diabetes remission seems to be highly arbitrary.

Response: Thank you for this important observation. While our study is retrospective, the tapering and discontinuation of OHAs were not arbitrary. As outlined in our previous response, deprescription was guided by a structured, principle-based framework (the RSSDI-ESI Therapeutic Wheel (https://doi.org/10.1007/s13410-020-00819-2)), which incorporates HbA1c levels, diabetes duration, weight loss, hypoglycemia risk, and comorbidities to support individualized treatment decisions. Although not rigidly protocolized, treating physicians followed an internal clinical algorithm that relied on daily self-monitored glycemic trends (fasting and postprandial values), clinical stability, and biochemical markers. Importantly, we acknowledge that clinical discretion and real-world variability may have influenced the timing and consistency of OHA withdrawal.

In line with your suggestion, we have revised the Limitations section (Page no.20, Lines 429 - 436) to reflect this, as follows:

“Further, the medication tapering followed a structured principle-based clinical protocol (RSSDI–ESI Therapeutic Wheel) and was guided by real-time glycemic trends, yet the retrospective nature of the study may have allowed for some variability in its implementation. Differences in physician judgment, documentation practices, and individual patient responses may have influenced the consistency of OHA withdrawal among participants. While this approach reflects real-world clinical practice, prospective studies with uniform deprescription algorithms and predefined remission criteria are warranted to validate these findings.”

3. Results, Fig. 2a and Fig. 3. Diabetes duration is analyzed by the cut-off of 6 years, while median diabetes duration is 7.6 years (line 202). Why the cut-off of 6 years is adopted?

Response: Thank you for your comment. The cut-off of 6 years for diabetes duration was chosen based on both clinical reasoning and prior research evidence. Studies such as the DiRECT trial have shown that individuals with shorter diabetes duration, particularly <6 years, are more likely to achieve remission due to better preserved beta-cell function and lower cumulative metabolic burden. Although the median diabetes duration in our cohort was 7.6 years, we opted for a 6-year threshold to reflect biologically meaningful stratification aligned with existing literature.

To further support this, we have clarified the rationale for this cut-off in the Material and Methods section as follows- (Page no.9, Lines 237 -240)

“For diabetes duration, a 6-year cut-off was applied based on existing evidence suggesting higher remission rates in patients with shorter disease duration [22]. This threshold enabled meaningful subgroup comparisons, despite the cohort’s median duration of 7.6 years.”

Minor comments

1. Eligibility criteria, line 100. Peripheral artery disease is not a part of neuropathy. Revise appropriately.

Response: Thank you for pointing this out. We acknowledge the error and have now revised the manuscript to list peripheral artery disease as a separate exclusion criterion, distinct from neuropathy. We have revised the section as follows - (Page no. 4, Lines 101 - 103)

“..significant neuropathy (including complete numbness or sensory loss, amputations, ulcers, foot deformities, or Charcot’s foot), or peripheral arterial disease..”

2. Table 1, line 213. Ref [15. 16] may be [16, 17]. Check the accuracy.

Response: Thank you for your comment. The discrepancy in numbering the references has been corrected. References for insulin cutoffs and HOMA-IR thresholds have been updated from “[15, 16]” to “[21, 23]”. (Page no. 10 , Line 259)

Reviewer 2

This study seeks to evaluate the remission of T2D in an Indian cohort using a culturally sensitive 4-component lifestyle intervention. Several concerns and questions are noted:

1. The statistical analysis section is poorly developed.

a. For the univariate logistic regression, the authors choose to convert continuous variables into categorical ones without any justification. The categorical cut-offs appear arbitrary, and the use of categories where continuous does not take full advantage of all the data available in the study. Moreover, why weren’t all the other variables in table 2 included in the logistic regression (e.g. weight, FBS)?

Response: Thank you for this important observation. Most of our continuous variables exhibited skewed distributions. Including them in their raw form would have required transformation (e.g., log-transformation), which could reduce interpretability and introduce complexity without materially improving model performance. Therefore, continuous variables were categorized using clinically relevant cut-offs (e.g., HbA1c <7%, weight loss ≥10%, HOMA-IR >2.5) to enhance the interpretability of results for clinical practice. We have now clarified this approach in the Materials and Methods section and added relevant references for the cut-offs used, as follows - (Page nos. 8 - 9 , Lines 235 -240)

“Continuous variables were categorized using clinically relevant thresholds (e.g., HbA1c <7% [19], weight loss ≥10% [20], and HOMA-IR >2.5 [21]) to enhance interpretability. For diabetes duration, a 6-year cut-off was applied based on existing evidence suggesting higher remission rates in patients with shorter disease duration [22]. This threshold enabled meaningful subgroup comparisons, despite the cohort’s median duration of 7.6 years.”

Regarding variable inclusion: weight change (≥10%) was included as a categorical predictor in the logistic regression (Fig 3). Other variables from Table 2, such as fasting blood glucose (FBG) were not included because they did not show significant bivariate associations with remission. We have clarified this in the statistical analysis section as follows- (Page no. 8, Lines 230 -234)

“The chi-square test was used to examine the associations between categorical variables. Variables showing significant bivariate associations with remission were included in the multivariable logistic regression analysis to identify independent predictors of diabetes remission (yes/no): age, HbA1c, medication status, weight change, HOMA-IR, juice fasting, intermittent fasting status, and disease duration.”

b. The generation and use of p-values in table 2 is confusing and distracting. Why are two different statistical tests being used for data with the same properties? Additionally, the authors reused the same data multiple times, but did not account for this in their analysis (i.e. no multiple comparison corrections).

Response: Thank you for your comment. Based on your suggestion, we have now incorporated Linear Mixed Model (LMM) analysis in Table 2 as the primary method to assess differences over time and between remission groups. The LMM accounts for repeated measures, inter-individual variability, and allows inclusion of covariates for adjustment. It also provides estimates for main time effects, group effects, and group × time interactions, offering a more robust and comprehensive analysis.

Additionally, Benjamini–Hochberg (BH) procedure is used to correct multiple test comparisons. This correction is now explicitly mentioned in the revised Materials and Methods section and table footnotes. Table 2 has been revised to present LMM-adjusted values clearly, and p-values are now derived solely from the LMM model. Separate significance tests for raw median values have been removed to reduce confusion. Additionally, effect sizes and 95% confidence intervals are reported alongside adjusted p-values to aid interpretation. (Page no. 12 -14 ).

In view of the additional analyses, we have revised the statistical analyses (Materials and Methods) and Results section as follows -

Materials and Methods (Page no.8, Lines 222 - 229 )

“Pre- and post-intervention comparisons, both for the total cohort and subgroup analyses, were conducted using a linear mixed-effects model (LMM) to account for repeated measurements and adjust for relevant covariates. To account for multiple comparisons across outcome variables, we applied the Benjamini–Hochberg (BH) procedure to control the false discovery rate at 5%. BH-adjusted p-values were computed separately for each set of comparisons (Time, Group, and Group × Time effects) in the LMMs. Adjusted p-values < 0.05 were considered statistically significant [18].”

Results (Page no.11, Lines 265 - 291)

“Post-intervention, 31.2% (n=744) of the participants achieved diabetes remission. Among drug-naïve participants, HbA1c improved significantly after the intervention [post-intervention: 6.0% (IQR: 5.6–6.3) vs. baseline: 6.9% (IQR: 6.5–7.5); p < 0.05], highlighting the effectiveness of lifestyle modification alone in improving glycemic control.

Table 2 provides a detailed comparison of anthropometric and biochemical parameters between the remission and non-remission groups at both the baseline and post-intervention, along with estimates from the LMM analysis. From baseline to post-intervention, the intervention resulted in significant improvements in weight, BMI, HbA1c, FBG, fasting insulin, and HOMA-IR across the entire cohort; no significant change was observed for HOMA-β.

Groupwise, significant differences were observed between the remission and no remission group, where at both time points, participants in the remission group consistently had lower weight, BMI, HbA1c, FBG, fasting insulin, HOMA-IR, and higher HOMA-β values (all p<0.05), indicating better glycemic and metabolic status.

Interaction analysis (Group x time) showed that participants in the remission group experienced a significantly greater reduction in weight, BMI, HbA1c, fasting insulin and HOMA-IR (all p<0.05). The changes in FBG and HOMA-β did not differ significantly between the groups (p>0.05). However, the relatively lower FBG levels and higher HOMA-β in the remission group suggested better glycemic control and preservation of β-cell function compared to the non-remission group.

Among the adjusted covariates, younger age was significantly associated with higher weight, BMI, FBG and HOMA-IR (all p<0.05). Regardless of time and remission status, females had lower weight and higher BMI than males (p<0.05), likely due to the height difference in the two sexes. Additionally, longer diabetes duration was associated with higher HbA1c and FBG levels and lower fasting insulin, HOMA-IR and HOMA-β (p<0.05).”

c. A multivariable logistic regression analysis should be performed.

Response: Thank you for your suggestion. We agree that a multivariable logistic regression model is appropriate for evaluating predictors of binary outcomes like remission. We would like to clarify that our analysis already used multivariable logistic regression, with multiple independent variables (e.g., age, weight loss, HbA1c, diabetes duration) included in the model to assess their association with remission. We have now updated the terminology in the Materials and Methods section to explicitly state this and avoid confusion.

We have revised the text as follows- (Page no.8, Lines: 231 - 234 )

“Variables showing significant bivariate associations with remission were included in the multivariable logistic regression analysis to identify independent predictors of diabetes remission (yes/no): age, HbA1c, medication status, weight change, HOMA-IR, juice fasting, intermittent fasting status, and disease duration.”

2. This authors state that this study was conducted at a Freedom from Diabetes Clinic using a 1 year online subscription-based model. It appears that the term post-intervention refers to the one year follow up period. There is no data presented that documents the length of time a patient was off OHAs at the one year mark nor how long drug naïve patients were found to be <6.5%. This should be included, as it is a central part of the study that is used to define remission.

Response: Thank you for your comment. We agree that the duration of OHA discontinuation and maintenance of HbA1c <6.5% is essential to substantiate remission status. We have now included additional data on both the duration of OHA-free status and HbA1c levels <6.5% among drug-naive participants under a dedicated heading in the Results section. The revised section reads as follows: (Page no.16, Lines 303 - 315)

“Sustainability of Glycemic Control and Remission Post-OHA Discontinuation

At one-year follow-up, 41.4% of participants had discontinued all OHAs and remained off-medication for a median duration of 9 months (IQR: 5.0–11.3). Overa

---

## [Decision Letter · Decision Letter 1]

10 Sep 2025

Type 2 diabetes remission and Its predictors in an Indian cohort: a retrospective analysis of an Intensive Lifestyle Intervention program

PONE-D-25-25515R1

Dear Dr. Kadam,

We’re pleased to inform you that your manuscript has been judged scientifically suitable for publication and will be formally accepted for publication once it meets all outstanding technical requirements.

Kind regards,

Guoying Wang, MD, PhD

Academic Editor

PLOS ONE

Additional Editor Comments (optional):

Reviewer #1:

Reviewer #2:

Reviewers' comments:

Reviewer's Responses to Questions

**Comments to the Author**

Reviewer #1: All comments have been addressed

Reviewer #2: All comments have been addressed

2. Is the manuscript technically sound, and do the data support the conclusions?

Reviewer #1: Yes

Reviewer #2: (No Response)

3. Has the statistical analysis been performed appropriately and rigorously?

Reviewer #1: Yes

Reviewer #2: (No Response)

4. Have the authors made all data underlying the findings in their manuscript fully available?

Reviewer #1: Yes

Reviewer #2: (No Response)

5. Is the manuscript presented in an intelligible fashion and written in standard English?

Reviewer #1: Yes

Reviewer #2: (No Response)

Reviewer #1: (No Response)

Reviewer #2: The manuscript has been improved and the authors provided thoughtful and appropriate reviewer responses.

**Do you want your identity to be public for this peer review?** For information about this choice, including consent withdrawal, please see our Privacy Policy

Reviewer #1: No

Reviewer #2: No

---

## [Editor Report · Acceptance letter]

PONE-D-25-25515R1

PLOS ONE

Dear Dr. Kadam,

I'm pleased to inform you that your manuscript has been deemed suitable for publication in PLOS ONE. Congratulations! Your manuscript is now being handed over to our production team.

Kind regards,

on behalf of

Dr. Guoying Wang

Academic Editor

PLOS ONE